# Estimating Mineral Requirements of Wild Herbivores: Modelling Arctic Caribou (*Rangifer tarandus granti*) in Summer

**DOI:** 10.3390/ani14060868

**Published:** 2024-03-12

**Authors:** Keith W. Oster, David D. Gustine, Fred E. Smeins, Perry S. Barboza

**Affiliations:** 1Alaska Department of Fish and Game, Division of Wildlife Conservation, P.O. Box 1467, Bethel, AK 99559, USA; keith.oster@alaska.gov; 2U.S. Fish and Wildlife Service, 1011 E. Tudor Road, Anchorage, AK 99503, USA; david_gustine@fws.gov; 3Department of Ecology and Conservation Biology, Texas A&M University, College Station, TX 77843, USA; f-smeins@tamu.edu; 4Department of Rangelands, Wildlife and Fisheries Management, Texas A&M University, College Station, TX 77843, USA

**Keywords:** factorial model, nutritional ecology, production, reproduction, ungulate

## Abstract

**Simple Summary:**

Wild herbivores require minerals to grow and maintain their skeleton and to catalyze their metabolic reactions. Herbivores acquire minerals from forage plants and, ultimately, the soils and waters that support seasonal plant growth. We know that seasonal changes in the quality and abundance of forage plants drive the movements of wild ungulates. However, the role of minerals in the movement and production of ungulates is poorly understood because the mineral requirements of wild species are poorly described. We describe a framework to estimate the minimum concentration of forage minerals required by female caribou, a wild ungulate with both migratory and sedentary ecotypes. We used measures of body mass and milk yield to determine the demand for minerals as the food intake changed with the season. We found that the minimum dietary concentrations (mg·kg^−1^) of macro-minerals (Ca, P, Mg, Na, K) declined as the food intake increased over the summer. The minimum dietary concentrations of micro-minerals (Fe, Mn, Cu, Zn) were heavily influenced by body mass gain, which increased through late lactation as food intakes rose. Our framework may be applied to other wild ungulates to assess the impacts of changing forage supplies on individuals and populations.

**Abstract:**

Mineral requirements are poorly described for most wildlife. Consequently, the role of forage minerals in movement and productivity are poorly understood for sedentary and migratory ungulates, such as reindeer and caribou (*Rangifer tarandus*). We applied estimates of maintenance, lactation, body mass change, and antler growth to production curves (body mass, daily intake, and milk yield) for female caribou to calculate their mineral requirements over summer. The total requirements (mg or g·d^−1^) were divided by the daily intake (kg·d^−1^) to estimate the minimum concentration of minerals required in the diet (mg or g·kg^−1^) to balance demand. The daily requirements (mg·d^−1^) of all minerals increased from parturition to the end of summer. The minimum dietary concentrations (mg·kg^−1^) of macro-minerals (Ca, P, Mg, Na, K) declined as food intake (kg·d^−1^) increased over summer. The minimum dietary concentrations (Fe, Mn, Cu, Zn) were heavily influenced by body mass gain, which increased through late lactation even though food intakes rose. Our modeling framework can be applied to other wild ungulates to assess the impacts of changing forage phenology, plant community compositions, or environmental disturbances on movement and productivity.

## 1. Introduction

Mammals require up to 29 minerals to maintain their bodies and to reproduce [1,2,3]. The structural demands for bones and homeostasis are greater than the demands for catalysis in metabolic pathways. Consequently, the demands for calcium (Ca), phosphorus (P), and magnesium (Mg) reflect the cost of maintaining bones and membranes even though all three elements are also involved in metabolic pathways. Daily demands for sodium (Na) and potassium (K) are driven by the maintenance of intracellular and extracellular exchanges of water and ions that affect both the integrity of the cells and their metabolic reactions. Daily exchanges of iron (Fe), manganese (Mn), copper (Cu), and zinc (Zn) are associated with proteins that act as carriers and catalysts. The requirements for these micro-minerals (Fe, Mn, Cu, and Zn) are expressed as intakes of mg·d^−^^1^ and at dietary concentrations in mg·kg^−^^1^. Macro-mineral (Ca, P, Mg, Na, and K) requirements are 1000 times greater and expressed as intakes of g·d^−^^1^ and at dietary concentrations of g·kg^−^^1^ [4].

The availability of mineral nutrients for herbivores varies with soil fertility and water availability, and with seasonal patterns in temperature and the light required for the growth of plants [5,6,7,8,9]. Mammalian herbivores obtain most of their minerals from the consumption of plants, but the direct consumption of soils (i.e., geophagy) can supplement the intake of forage minerals in some regions [10,11,12,13]. Seasonal plant growth influences the movement of ungulates that must forage widely to meet the demands of their large body size over several cycles of plant growth through their long lives [14]. Migrations of wildebeests (*Connochaetes taurinus*) in Africa [15] are associated with changes in the availability of water and P, while variation in Na is associated with the movements of North American ungulates [12,16]. It is difficult to assess the direct effects of changing mineral availabilities on the movement and production of wild ungulates because mineral requirements have not been quantified for most wild species [12,17]. However, the requirements for wild ungulates may be estimated from those of domestic ungulates by accounting for differences in seasonal food intake, mass gain, and reproduction [18].

The movement and production of free-ranging ungulates affect human livelihoods and ecological processes across grassland, savanna, forest, and tundra communities [19,20,21]. Caribou and reindeer have a circumpolar distribution, where winters are severe, and the growth of forage is limited to a brief summer. Female caribou migrate over 3000 km each year to complete an annual cycle that begins with their migration to calving grounds and the emergence of new vegetation in spring [22,23]. Migratory caribou and sedentary reindeer rely on the spring growing season to replenish nutrient stores lost during the preceding winter and to support the high costs of lactation and body mass gain over the summer [24]. Females that fail to store adequate nutrients during the summer may not breed the following year and may be vulnerable to winter mortality [25]. The rise and fall in plant quality during summer creates a window of opportunity to store fat and body protein when the digestible energy and digestible protein in the forage exceed a threshold for growth or reproduction [26,27,28]. Similarly, windows of opportunity emerge for gaining body minerals as plants grow and senesce over the summer [7]. Forage concentrations of Ca, P, and Fe are affected by differences in soil acidity among the migratory ranges of caribou in Arctic Alaska [7]. Similarly, the proximity to the ocean affects the availability of Na in caribou forage, especially in spring, when the forage K is highest. Caribou therefore face the challenge of retaining enough of one mineral with a low forage content (e.g., Cu) as the concentration of a competing mineral (e.g., Zn) increases with the growth of the plant over summer [7]. The mineral status of caribou at the end of the summer is therefore dependent on the timing of mineral availability in the landscape, the timing of nutrient requirements over the summer, and the ability of caribou to store those minerals in body tissues.

We used a factorial approach to derive the mineral requirements of lactating female caribou based on the best available information. We combined measures of intake and body mass change in captive caribou [27] with the published requirements for domestic cattle (*Bos taurus*), goats (*Capra hircus*), horses (*Equus ferus caballus*), and sheep (*Ovis aries*) (Figure 1). Our factorial estimates use the requirements for maintenance (i.e., survival) and production (i.e., body mass gain and reproduction). The observed metrics of individual production for caribou (mass change, food intake, milk production, and antler synthesis) were compiled from the literature and applied to the factorial relationships of the mineral requirements of domestic animals. Models of nutrient requirements in wild ungulates traditionally focus on their requirements for protein and energy by using the consensus of the literature regarding specific guilds of ruminants [29] or the functional relationships between net primary production, body size, and population size [30], or by incorporating data on wild ungulates into the factorial requirements of domestic ruminants [18]. Our model is the first to estimate minerals requirements by incorporating seasonal patterns in the food intake and body mass gain of a wild ungulate with factorial estimates of mineral requirements in domestic ungulates.

## 2. Materials and Methods

Daily dry matter intake (DMI) and body weight (BW) were predicted as a function of average daily air temperature and the number of days from calving for wild-caught, captive caribou in Fairbanks, Alaska (65° N 146° W) [27]. This model was derived from individual measures of food intake and body mass for 10 parturient caribou consuming rations formulated to mimic the low energy density of sedges (digestible energy content of 10.8 kJ·g^−1^ of dry mass) [27]. We simulated food intake and body mass of female caribou in Alaska from parturition in spring to the start of the mating season in early fall (120 d). Air temperature (T) was simulated from day of parturition (B) as
T = (0.1680286·B) + (−0.0005179·B^2^) + (−8.24·10^−6^·B^3^) + (10.16155). (1)

Mass specific intake of dry food (DMI g·kg^−^^0.75^·d^−^^1^) was calculated as
DMI = (5.576509·B) + (−0.0726042·B^2^) + (0.0002894·B^3^) + (−2.692504·T) + (124.7071).(2)

Body mass (BW kg) was calculated from day of parturition as
BW = (−0.1181081·B) + (0.0019413·B^2^) + 92.07143.(3)

Food intake was calculated from DMI and BM on a daily time step. Mass change was calculated at a daily time step from the simulated body mass. Milk yield (MY) was derived from the curve of Parker et al. (1990) [31] to achieve a peak yield of 1.9 kg/d at 30 days from birth using the following expression in SigmaPlot (version 12.0; Systat Software Inc., San Jose, CA, USA): (4)$$MY=\text{1.1081}+\text{27.2543}\times \frac{e^{-\text{0.5}\times \left(\frac{\ln\left(\frac{B}{\text{37.7865}}\right)}{\text{0.4872}}\right)}}{B}$$

We used a factorial approach to calculate total intake of minerals required to meet demands for maintenance, as well as body mass change, lactation, and antler growth (Figure 1). Mineral requirements of female caribou were simulated with relationships that were derived for goats, sheep, dairy cattle, beef cattle, and horses at both maintenance and production for Ca, P, Mg, Na, K, Mn, Fe, Cu, and Zn (Table 1 and Table 2) [32,33,34,35]. When mass specific requirements for a mineral were not available, we compared our factorial estimates to feeding standards that provide a safe and adequate mineral intake or dietary concentration [32]. 

Published requirements of minerals for maintenance were based on food intake (dry matter basis) and body mass (Table 1 and Table 2). Model caribou were 92 kg at parturition and 106 kg at the end of lactation (Figure 2). Model caribou increased food intakes over the first 50 d from parturition. High air temperatures coincided with a plateau in intake mid lactation, which was followed by another increase in food intake from 90 to 120 days from parturition (Figure 2). Daily mass change of model caribou increased in a linear fashion from −116 to +346 g·d^−^^1^ through the summer. Factorial requirements of minerals for body mass change were based on daily body mass and change in mass (i.e., live weight gain) in relation to a target mass (i.e., mature mass) (Table 1 and Table 2). We assumed that estimates of mineral requirements for growth were applicable to both body mass gain and body mass loss in adult female caribou. Minerals released from mobilization of body mass were therefore completely used for mineral demands that in turn reduced the mineral demand from the diet. We used mineral concentration of reindeer milk to estimate requirements for lactation in all model projections as a product of the caribou milk yield curve (Figure 2) [36].

We calculated the deposition of dry antler at a linear rate from day of calving to a final antler mass of 375 g, which was 0.34% of the maximum body mass (110 kg) attained in December [37]. Dry antler composition was calculated as containing 70% ash when combusted at 500 °C for 4 h [27]. Antler ash was then assumed to be composed of pure hydroxyapatite with hydrogen and oxygen components volatilized during combustion (68% Ca, 32% P) [38]. Requirements for maintenance and growth of body mass, lactation and antler growth were summed and divided by an absorption coefficient (AC) for each projection to determine the daily requirement of dietary minerals. Daily requirements were divided by the daily food intake to project the minimum concentration of minerals in the diet (mg·kg^−1^ dry mass) required to achieve zero balance (Figure 1). Early lactation requirements in caribou were calculated as the mean of all projections from 1 to 28 d of lactation, while late lactation requirements were calculated from 29 to 120 d of lactation. We used ordinary least squares regression of minimum dietary mineral concentration (Y) against time from birth (X) to describe the dietary requirement of each mineral for caribou as a cubic polynomial function (STATA version 14.1; Stata Corp., College Station, TX, USA). 

## 3. Results

### 3.1. Calcium, Phosphorus, and Magnesium

The projections of the daily requirements for dietary Ca at maintenance increased over the summer and ranged from 4.2 to 10.8 g Ca·d^−^^1^. The lactation demands for dietary Ca added 7.1–20.4 g·d^−^^1^ to daily mineral demands. The mobilization of Ca from body tissue in early lactation reduced dietary demand by 1.9 g·d^−^^1^, whereas 11.5 g·d^−^^1^ was needed from the diet to deposit Ca in tissue at the end of summer. The daily requirements for Ca for antler growth were constant over time and ranged from 2.98 to 3.5 g Ca·d^−^^1^ among species projections. The total daily requirements for dietary Ca increased throughout the summer and were lowest for projections from the sheep model (12.7–24.5 g Ca·d^−^^1^) and highest for projections from the dairy cattle model (16.1–32.4 g Ca·d^−^^1^). Our estimate of minimum dietary Ca concentration decreased from days 1 to 80 (4694 to 3917 mg Ca·kg^−^^1^) as intakes increased (Table 3). Increases in the minimum concentration of dietary Ca during late lactation were associated with demands for mass gain (Figure 3). Consequently, the estimates of minimum dietary concentration of Ca were not significantly different between early and late lactation (Table 4).

The projections of the daily dietary requirements of P for maintenance increased over the summer and ranged from 4.2 to 13.4 g P·d^−^^1^. The projected dietary requirements of P for lactation in addition to maintenance ranged from 4.4 to 14.8 g P·d^−^^1^. The mobilization of P from body tissue reduced dietary demand by 0.9 g P·d^−^^1^ during early lactation, while 7.9 g·d^−^^1^ of dietary P was required to support tissue gain at the end of summer. The daily P requirements for antler growth ranged from 1.1 to 2.0 g P·d^−1^ among models. The total daily requirements for dietary P increased through the summer and were lowest in the projections of the goat model (9.3–19.0 g P·d^−1^) and greatest in the projections of the horse model (17.0–27.0 g P·d^−1^). The estimates for the minimum dietary concentration required to achieve zero balance in caribou decreased from day 1 to 95 (3721 to 3119 mg P·kg^−1^) as intakes increased (Table 3). Increases in dietary concentration after day 95 were associated with the demands of mass gain (Figure 3). As a result, the required dietary concentrations of P were not significantly different between early and late lactation (Table 4).

The daily requirements of dietary Mg for maintenance increased over the summer and ranged from 1.6 g to 3.2 g Mg·d^−^^1^. The minimum and maximum projections for lactation in addition to maintenance ranged from 0.4 to 2.3 g Mg·d^−1^. The body mass lost at the start of lactation reduced the dietary demands for Mg by 0.3 g·d^−^^1^, while the mass gained at the end of the summer increased requirements by 1.0 g Mg·d^−^^1^. The absolute daily dietary requirements for Mg increased throughout the summer and were lowest in the projections of the beef cattle model (2.4–3.4 g Mg·d^−1^) and highest in the projection of the horse model (3.0–4.5 g Mg·d^−1^). Our estimates of the minimum dietary concentration of Mg decreased from day 1 to 60 (819 to 537 mg Mg·kg^−1^) and remained at 537 to 554 mg Mg·kg^−1^ (Figure 3, Table 3). Consequently, the required dietary concentrations of Mg were significantly greater in early lactation than in late lactation (Table 4).

### 3.2. Sodium and Potassium 

The daily Na requirements for maintenance increased over the summer from 0.7 to 4.5 g Na·d^−^^1^. Lactation increased the demands for dietary Na by 0.7–1.4 g·d^−^^1^. Mass loss reduced the dietary demand of Na by 0.2 g·d^−^^1^ at the start of lactation, whereas the mass gain at the end of summer increased Na demands by 0.7 g·d^−^^1^. The daily requirements for total dietary sodium increased throughout the summer and were lowest in the projections of the sheep model (1.7–2.4 g Na· d^−1^) and greatest in the projections of the dairy cattle model (4.5–5.8 g Na·d^−1^). The minimum dietary concentrations of required Na decreased from day 1 to 60 (827 to 512 mg Na·kg^−1^) and remained at 520–540 mg Na·kg^−1^ until the end of summer (Figure 4, Table 3), but the dietary minima for Na were not significantly different between early and late lactation (Table 4).

The projected K requirements for maintenance differed significantly between model ruminants and horses: the projections for the horse were only 5.8–6.8 g K·d^−^^1^, whereas the projections for the maintenance of sheep, goats, and dairy cattle were 12.3–27.3 g K·d^−^^1^. Lactation increased the demands for dietary K from 1.9 to 4.0 g K·d^−^^1^ among species projections. Tissue mobilization in early lactation reduced dietary demands by 0.3 g K·d^−^^1^, while the deposition of body tissue at the end of summer increased dietary demand by 0.9 g K·d^−1^. The daily requirements for dietary K increased through the summer and were lowest in the projections of the horse model (8.0–9.8 g K·d^−1^) but were comparable in projections of sheep, goat, and dairy cattle models (14.4–28.4 g K·d^−1^). The minimum dietary K concentration decreased from day 1 to 80 (4081 to 3278 mg K·kg ^−1^) and remained at 3279–3336 mg K·kg^−1^ through late lactation (Figure 4, Table 3). 

### 3.3. Iron, Manganese, Copper, and Zinc

The projections of daily maintenance requirements for Fe were only based on the models for sheep and dairy cattle, because the requirements for beef cattle, goats, and horses were unavailable from the National Research Council. The maintenance requirements were 12.6 to 14.8 mg Fe·d^−^^1^, with an additional 0.8–0.13 mg Fe·d^−^^1^ for lactation. The body mass lost in early lactation reduced dietary demands for Fe by 6.4 mg Fe· d^−^^1^, while mass gained at the end of summer increased the dietary demand by 190 mg Fe required·d^−^^1^. The total daily requirements increased over the lactation period and were (6.6–205.1 mg Fe·d^−^^1^) in sheep model projections and (0–117.7 mg Fe·d^−^^1^) in dairy cattle model projections. The minimum dietary concentrations of Fe required in the diet were only 0.6–1.5 mg Fe·kg^−1^ from day 1 to 20 but increased through lactation to 21.7 mg Fe·kg^−1^ (Figure 5, Table 3).

The daily maintenance requirements for Mn increased over the summer (18.1 to 28.8 mg Mn·d^−^^1^), with an additional 0.09–0.20 mg Mn·d^−^^1^ required for lactation. The body mass lost at early lactation reduced dietary demands by 0.08 mg Mn·d^−^^1^, while mass gained at the end of summer increased dietary demands by 32.3 mg Mn·d^−^^1^. The total daily requirements of Mn increased through the summer and were lowest in the projections of the dairy cattle model (3.2–6.2 mg Mn·d^−^^1^) and highest in the projections of the goat model (4.2–8.2 mg Mn·d^−^^1^). The minimum dietary concentrations of Mn declined from 6.8 to 4.0 mg Mn·kg^−^^1^ until day 40 but increased to 7.0 mg Mn·kg^−^^1^ by day 110 of lactation (Figure 5, Table 3). 

The daily maintenance requirements for Cu increased over summer (6.0 to 20.9 mg Cu·d^−^^1^), with only an additional 0.003–0.04 mg Cu·d^−^^1^ required for lactation. The body mass lost reduced dietary demands in early lactation by 0.13 mg Cu·d^−^^1^, while the body mass gained at the end of summer only increased the dietary demands for Cu by up to 10 mg Cu·d^−^^1^. The daily demands for copper increased through the summer and were lowest in the projections of the sheep model (6.0–13.2 mg Cu·d^−^^1^) and highest in the projections of the horse model (16.0–28.8 mg Cu·d^−^^1^). The minimum dietary concentrations of Cu required in the diet declined from 4.0 to 2.2 mg Cu·kg^−1^ to day 50 and increased to 3.0 mg Cu·kg^−1^ through late lactation (Figure 5, Table 3).

The maintenance requirements for Zn increased over the summer (27.1 to 53.6 mg Zn·d^−^^1^ among models), with an additional 8.3–14.4 mg Zn·d^−^^1^ required for lactation. The body mass lost in early lactation reduced the dietary demand by 2.9 mg Zn·d^−^^1^, while the body mass gained at the end of summer increased the dietary demand by 57.6 mg Zn·d^−^^1^. Daily demands for Zn increased through the summer and were lowest in the projections of the dairy cattle model (33.1–95.5 mg Zn·d^−^^1^) and highest in the projection of the sheep model (52.2–117.4 mg Zn·d^−1^). The minimum dietary concentrations of Zn decreased from 11.9 to 8.4 mg Zn·kg^−1^ over days 1–40 and rose to 14.0 mg Zn·kg^−1^ through late lactation (Figure 5, Table 3). 

## 4. Discussion

We used a consensus approach to estimate the mineral requirements of caribou by applying the food intake and body mass curves of captive caribou to the factorial requirements of domestic ungulates that were compiled from the literature by the National Research Council [32,33,34,35]. We found that the total requirements for minerals (g·d^−1^) are greater in late lactation than in early lactation, even though the digestible contents of energy and protein in forages are greatest in early lactation [26,39]. The maintenance costs of minerals (g·d^−1^) increased throughout the summer due to increased fecal and urinary excretion as the both intake and body mass increased. The daily mineral demands for lactation (g·d^−^^1^) were highest at peak lactation but similar at the start and end of the lactation period. The mass loss in early lactation reduced the demand for minerals from the diet (g·d^−^^1^), but the restoration of body mass at the end of summer greatly increased demands, because these elements were absorbed at <20% efficiency (Table 2). The costs of maintenance and/or mass gain outweighed the demands for lactation such that the daily mineral demands for all minerals in our study were highest at the end of summer. Caribou rely on stores of body protein for winter pregnancy and spring lactation that are established at the end of the prior summer [27,40]. Our models indicate that mineral stores in lean mass may also be established in late summer and early winter. The importance of Ca and P stores for caribou is supported by the prevalence of osteophagia in spring—60% of antlers shed on their calving grounds are marked by the teeth of caribou [41] 

Our projection of daily food intake had a large influence on the estimate of minimum required dietary concentration (mg·kg^−1^). High demands at low intakes in early lactation elevated the minima, whereas high intakes in late lactation lower the estimated minimum [27,42]. The dietary minima (mg·kg^−1^) of macrominerals (Ca, P, Mg, Na, and K) were high in early lactation due to low intakes early in the summer but declined through peak lactation as intakes increased (Figure 2, Figure 3 and Figure 4). Conversely, the dietary minima for trace minerals such as Fe, Mn, Cu, and Zn (mg·kg^−1^) increased in late lactation with increasing demands for tissue gain even though food intakes also increased during this period (Figure 5). Warm summer days may increase the minimum dietary concentrations of minerals for female caribou by depressing food intake (Figure 2) [27]. Heat waves on consecutive warm days in mid-summer reduce the foraging activity and body mass gains of female reindeer [43]. Warm days are also associated with increased insect harassment in arctic caribou that also affects movement and foraging [39].

Our estimates of mineral demands include those for growing domestic animals that are probably higher than those required for the seasonal regain of fat and lean mass in adult animals, such as reproductive caribou (Table 1). The allometric effects on our estimates were probably small because we used body mass relationships to predict the demands of a species at ~100 kg, which is within the distribution of body masses of sheep, goats, cattle, and horses in the source data. The estimated requirements of Ca, P, and Mg (g·d^−^^1^) for mass gain in caribou may be overestimated because the factor is partly based on young animals that are building bones with high concentrations of these minerals. Trace minerals that are incorporated into the lean mass of soft tissues may be less influenced by our bias toward young animals. However, the development of mineral stores may differ among species. For example, Cu is deposited in the liver of young caribou and muskoxen during late lactation as they progress toward weaning when demands for Cu may be further increased by the immune response to infections [44,45].

Absorption efficiency has a strong effect on the estimate of mineral requirements when animals are gaining mass (Table 1 and Table 2). Absorption efficiencies vary with the mineral, the physiological state of the animal, and interactions among minerals in the diet. The bioavailable inorganic mineral additives used in formulated rations tend to be better absorbed than minerals in natural forages [1]. Mineral absorption from natural forages also changes with plant phenology, as minerals become less digestible as forages approach senescence [5]. True absorption coefficients have been traditionally determined by feeding captive animals formulated diets, with mineral concentrations below the minimum requirements, in order to fully activate physiological absorption mechanisms [34]. While absorption coefficients for natural forages were used when available, most models of mineral requirements used coefficients from domestic animals on formulated rations with a marginal mineral balance. As a result, our model projections likely overestimate the true absorption efficiency in caribou on a natural forage diet. 

Our estimates of the minimum dietary concentrations of Ca, P, Mg, Na, and K for lactating caribou (Table 4) are within the range of concentrations reported for forages used by caribou in Alaska during summer [46,47]. Furthermore, our estimates of the minimal concentrations of Ca and P required for caribou are similar to those derived for cervids by the National Research Council [32]. We applied the National Research Council’s recommended requirements for Ca and P for cervids to our model of caribou production. We found that the projected Ca demand for cervids was higher than those projected by our models (Table 4; 7476 to 4956 mg Ca·kg^−1^). Conversely, the cervid projection for P was lower than those projected using our models (Table 4; 3960 to 2248 mg P·kg^−1^). Staaland and White [16] estimated the dietary concentrations of Ca, P, Mg, Na, K, Mn, and Cu required in the diet of caribou at peak lactation based on data for lactating sheep (2 L·d^−1^ milk yield). Our estimates in Table 4 are 27–75% greater than those reported by Staaland and White [16] for macro-minerals (2685 mg Ca·kg^−1^; 2694 mg P·kg^−1^, 505 mg Na·kg^−1^, and 3206 mg Na·kg ^−1^) but only 30–80% of the reported concentrations for Mg, Mn, and Cu (1507 mg Mg·kg^−1^, 22 mg Mn·kg^−1^, and 5 mg Cu·kg^−1^). Discrepancies between estimated requirements are likely due to our estimates including measures of antler growth and mass gain that were omitted from the estimates of Staaland and White [16]. In addition, we used concentrations of Ca, P, Na, and K for reindeer milk that are higher than those for sheep. Conversely, the concentrations of Mg, Mn, and Cu in sheep milk [32] are higher than those reported for reindeer milk and may explain the lower estimates of dietary minima derived in our models.

## 5. Conclusions

Our framework may be used to evaluate the response of caribou and reindeer to changes in their nutrient supply within their seasonal range or among their migratory ranges. A similar approach could be used with multiple species to assess the effects of changing forage resources and environmental demands on the movement and production of domestic and wild ungulates in rangelands through products such as the Rangeland Analysis Platform [48,49]. However, further applications of our model relationships should be verified using direct measures of wild caribou with markers of mineral status, such as bone density (Ca, P), metalloenzymes (Cu, Zn, Fe), and tissue concentration (e.g., liver Cu and Zn) in relation to seasonal movement and diet [50,51]. 

Our framework may also be used to assess the contributions of caribou to elemental cycles and changes in plant communities. Caribou and reindeer shape local communities by removing plants through consumption and trampling and through the deposition of minerals and N in urine and feces that facilitate plant growth [52,53]. The large-scale movements of caribou translocate elements when they lose antlers, or lose their lives to predators, disease, and exposure. Antlers dropped on calving grounds have the isotopic signatures of elements in the summer range where those antlers were grown [54]. The role of caribou in the arctic ecosystem has increased with cycles of glaciation that favored species suited to variable forage supplies [55]. Recent arctic warming is altering plant communities and geochemical processes that are likely to continue altering the role of caribou in the ecosystem [56,57,58].

## Figures and Tables

**Figure 1 animals-14-00868-f001:**
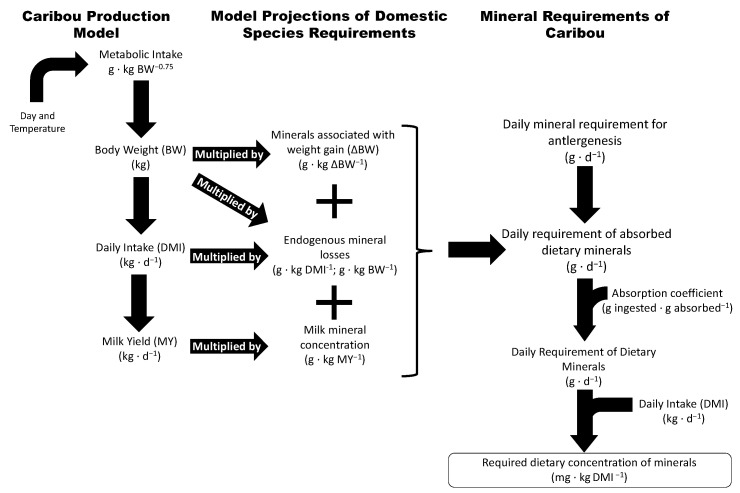
Flow diagram for estimating the mineral requirements and the minimum dietary concentrations for lactating female caribou in three stages: daily attributes of caribou; specific requirements; and projected daily requirements for minerals.

**Figure 2 animals-14-00868-f002:**
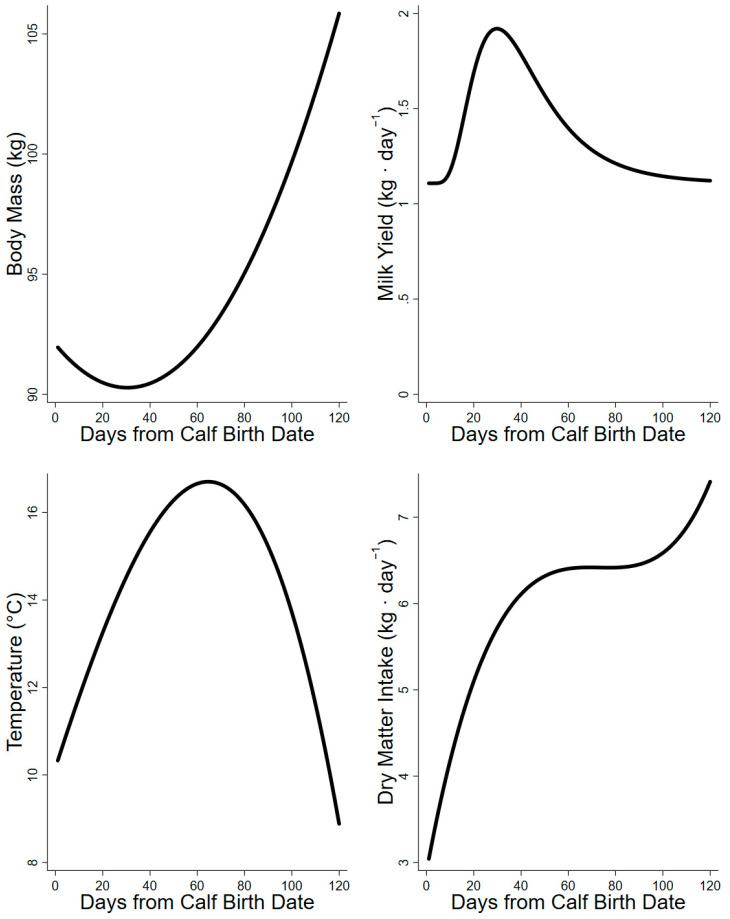
Model inputs for estimating requirements of female caribou from parturition through 120 days of lactation. Body mass (kg), milk yields (kg·d^−^^1^), temperature (°C), and dry matter intake (kg·d^−^^1^) were derived from measures of captive caribou in interior Alaska.

**Figure 3 animals-14-00868-f003:**
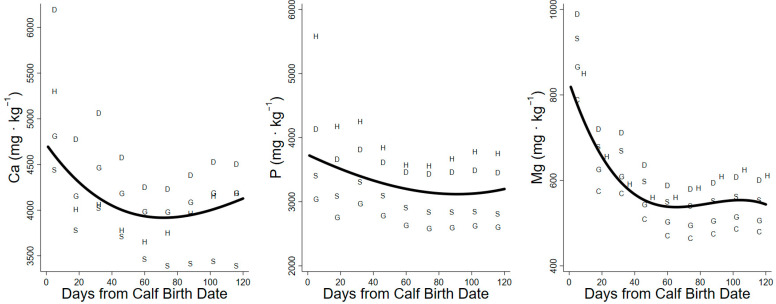
Minimum dietary concentration (dry matter basis) of Ca, P, and Mg calculated for lactating caribou with factorial relationships for domestic ungulates. S = sheep model projection, G = goat model projection, D = dairy cattle model projection, C = beef cattle model projection, H = horse model projection. Solid lines represent the regression of all species projections used for each element.

**Figure 4 animals-14-00868-f004:**
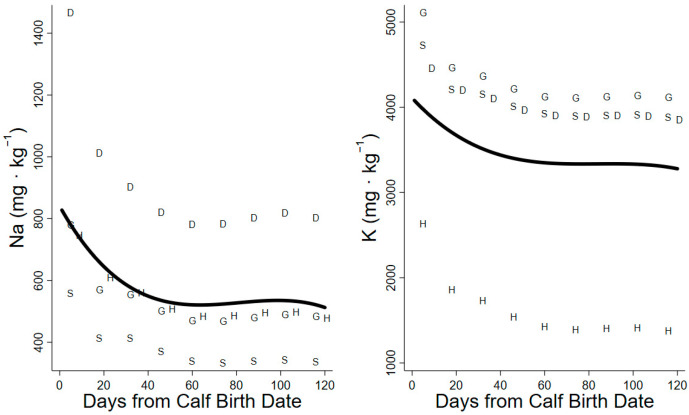
Minimum dietary concentration (dry matter basis) of Na and K calculated for lactating caribou with factorial relationships for domestic ungulates. S = sheep model projection, G = goat model projection, D = dairy cattle model projection, H = horse model projection. Solid line represents the regression of species projections used for each element.

**Figure 5 animals-14-00868-f005:**
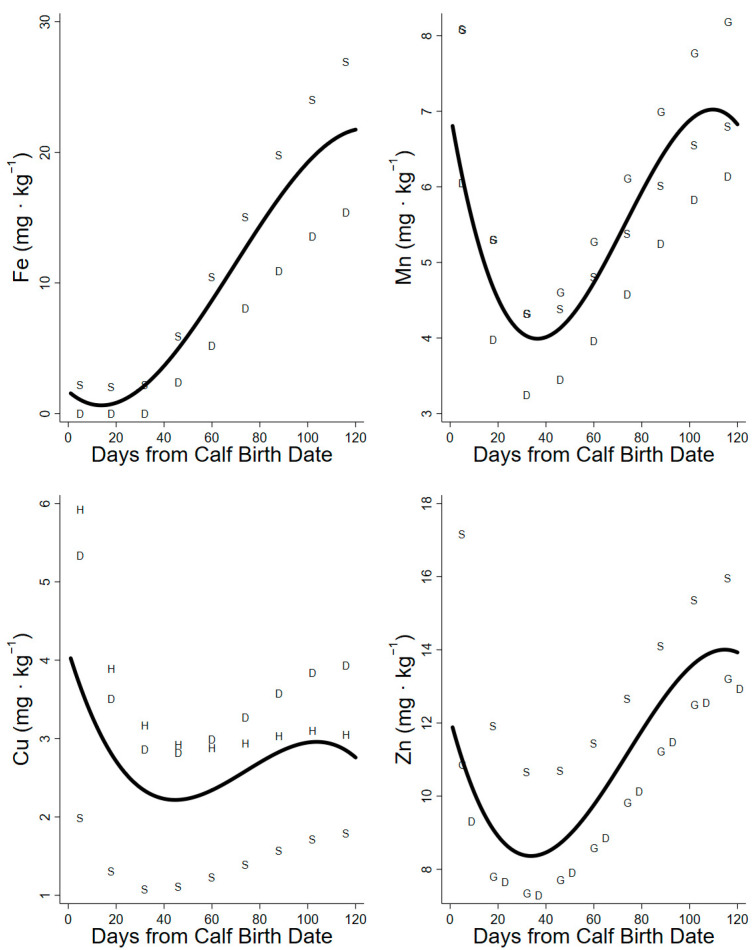
Minimum dietary concentration (dry matter basis) of Fe, Mn, Cu, and Zn calculated for lactating caribou with factorial relationships for domestic ungulates. S = sheep model projection, G = goat model projection, D = dairy cattle model projection, H = horse model projection. Solid line represents the regression of species projections used for each element.

**Table 1 animals-14-00868-t001:** Factorial relationships for requirements of Ca, P, Mg, Na, and K in sheep, goats, beef cattle, dairy cattle, and horses. BW = body weight (kg), MW = mature weight (kg), DMI = dry matter intake (kg), LWG = live weight gain (kg), LMG = lean mass gain (kg), MY = milk yield (kg), AC = absorption coefficient.

Mineral	Taxa	AC (g·g^−1^)	Maintenance (g·d^−1^)	Body Mass Change (g·d^−1^)	Lactation (g·d^−1^)
Ca	Sheep	0.5	(0.623·DMI^−1^) + 0.228	LWG·6.25·MW^0.28^·BW^−0.28^	3.2·MY^−1^
	Goats	0.45	(0.623·DMI^−1^) + 0.228	11·LWG^−1^	3.2·MY^−1^
	Beef cattle	0.5	0.0154·BW^−1^	^a^ 71·LMG^−1^	3.2·MY^−1^
	Dairy cattle	0.3	0.0154·BW^−1^	9.83·(MW^0.22^)·(BW^−0.22^))·LWG	3.2·MY^−1^
	Horses	0.5	0.043·BW^−1^	16·LWG^−1^	3.2·MY^−1^
P	Sheep	0.6	1.6·(0.603·DMI − 0.06)	LWG·(1.2 + 3.188·MW^0.28^·BW^−0.28^)	2.7·MY^−1^
	Goats	0.65	0.081 + (0.88·DMI^−1^)	5.8·LWG^−1^	2.7·MY^−1^
	Beef cattle	0.64	0.016·BW^−1^	^a^ 39·LMG^−1^	2.7·MY^−1^
	Dairy cattle	0.68	(1.2·DMI^−1^) + (0.002·BW^−1^)	1.2 + (4.635·MW^0.22^)·(BW^−0.22^)·LWG	2.7·MY^−1^
	Horses	0.35	0.028·BW^−1^	8.0·LWG^−1^	2.7·MY^−1^
Mg	Sheep	0.17	0.003·BW^−1^	0.41·LWG^−1^	0.193·MY^−1^
	Goats	0.2	0.0035·BW^−1^	0.4·LWG^−1^	0.193·MY^−1^
	Beef cattle	0.2	0.003·BW^−1^	0.45·LWG^−1^	0.193·MY^−1^
	Dairy cattle	0.16	0.003·BW^−1^	0.45·LWG^−1^	0.193·MY^−1^
	Horses	0.5	0.0015·BW^−1^	1.25·LWG^−1^	0.193·MY^−1^
Na	Sheep	0.91	0.0108·BW^−1^	1.1·LWG^−1^	0.6·MY^−1^
	Goats	0.8	0.015·BW^−1^	1.6·LWG^−1^	0.6·MY^−1^
	Dairy cattle	0.9	0.038·BW^−1^	1.4·LWG^−1^	0.6·MY^−1^
	Horses	0.9	0.02·BW^−1^	1.0·LWG^−1^	0.6·MY^−1^
K	Sheep	0.9	(2.6 g·DMI^−1^) + (0.038 g·BW^−1^)	1.8·LWG^−1^	1.56·MY^−1^
	Goats	0.9	(2.6 g·DMI^−1^) + (0.05 g·BW^−1^)	2.4·LWG^−1^	1.56·MY^−1^
	Dairy cattle	0.9	(2.6·DMI^−1^) + (0.038 g·BW^−1^)	1.6·LWG^−1^	1.56·MY^−1^
	Horses	0.75	0.048·BW^−1^	1.5·LWG^−1^	1.56·MY^−1^

^a^ Not representative of mass gain in adult animals.

**Table 2 animals-14-00868-t002:** Factorial relationships for requirements of Fe, Mn, Cu, and Zn in sheep, goats, beef cattle, dairy cattle, and horses. BW = body weight (kg), MW = mature weight (kg), DMI = dry matter intake (kg), LWG = live weight gain (kg), LMG = lean mass gain (kg), MY = milk yield (kg), AC = absorption coefficient.

Mineral	Taxa	AC (g·g^−1^)	Maintenance (mg·d^−1^)	Body Mass Change (mg·d^−1^)	Lactation (mg·d^−1^)
Fe	Sheep	0.1	0.014·BW^−1^	55·LWG^−1^	0.007·MY^−1^
	Dairy cattle	0.1	0·BW^−1^	34·LWG^−1^	0.007·MY^−1^
Mn	Sheep	0.0075	0.002·BW^−1^	0.47·LWG^−1^	0.0008·MY^−1^
	Goats	0.0075	0.002·BW^−1^	0.7·LWG^−1^	0.0008·MY^−1^
	Dairy cattle	0.01	0.002·BW^−1^	0.7 mg·LWG^−1^	0.0008·MY^−1^
Cu	Sheep	0.06	0.004·BW^−1^	1.06·LWG^−1^	0.0008·MY^−1^
	Dairy cattle	0.04	0.0071·BW^−1^	1.15·LWG^−1^	0.0008·MY^−1^
	Horses	0.35	0.069·BW^−1^	1·LWG^−1^	0.0008·MY^−1^
Zn	Sheep	0.15	0.076·BW^−1^	24·LWG^−1^	1.13·MY^−1^
	Goats	0.15	0.045·BW^−1^	25·LWG^−1^	1.13·MY^−1^
	Dairy cattle	0.15	0.045·BW^−1^	24·LWG^−1^	1.13·MY^−1^

**Table 3 animals-14-00868-t003:** Minimum dietary concentrations of minerals (Y mg · kg^−1^ of Ca, P, Mg, Na, K, Fe, Mn, Cu, or Zn) required by in female caribou in relation to time from birth (x; days).

Mineral	Requirement Phenology Regression	Adjusted r^2^
Ca	Y = −25.32x + 0.2375x^2^ − 0.00056x^3^ + 4719.26	0.20
P	Y = −11.92x + 0.0399x^2^ + 0.00019x^3^ + 3732.65	0.11
Mg	Y = −11.31x + 0.1407x^2^ − 0.00055x^3^ + 829.89	0.64
Na	Y = −12.8x + 0.1656x^2^ − 0.00068x^3^ + 840.15	0.14
K	Y = −28.08x + 0.3377x^2^ − 0.00134x^3^ + 4108.27	0.03
Fe	Y = −0.16x + 0.0065x^2^ − 0.00003x^3^ + 1.7	0.81
Mn	Y = −0.19x + 0.0034x^2^ − 0.00002x^3^ + 6.99	0.71
Cu	Y = −0.1x + 0.0016x^2^ − 0.00001x^3^ + 4.12	0.14
Zn	Y = −0.25x + 0.0047x^2^ − 0.00002x^3^ + 12.13	0.61

**Table 4 animals-14-00868-t004:** Estimated requirements of female caribou (mean ± SD) from birth through 120 days of lactation based on factorial relationships for selected domestic ungulates. Estimates for early lactation are averaged from birth to peak lactation (28 days from parturition). Estimates for late lactation are averaged from 29 to 120 days from parturition. S = sheep model projection, G = goat model projection, D = dairy cattle model projection, C = beef cattle model projection, H = horse model projection.

Mineral	Incorporated Models	Peak Dietary Requirement (mg·kg DMI^−1^)	Day of Peak Requirement	Early Lactation Requirement (mg·kg DMI^−1^ ± SD)	Late Lactation Requirement (mg·kg DMI^−1^ ± SD)
Ca	S,G,D,H	4694	120	4406 ± 484	3999 ± 397
P	S,G,D,H	3721	1	3567 ± 696	3202 ± 499
Mg	S,G,C,D,H	819	1	719 ± 42	570 ± 40
Na	S,G,D,H	828	1	691 ± 267	535 ± 194
K	S,G,D,H	4653	1	3775 ± 1213	3364 ± 1285
Fe	S,D	21.74	120	1.03 ± 1.46	12.43 ± 5.26
Mn	S,G,D	7.02	110	5.10 ± 0.81	5.56 ± 0.79
Cu	S,D,H	4.02	1	3.04 ± 1.46	2.60 ± 1.02
Zn	S,G,D	14	115	9.67 ± 2.48	11.22 ± 1.72

## Data Availability

The food intake and body mass of caribou are described by Barboza et al. [27] and are available at the Texas Data Repository https://doi.org/10.18738/T8/UIRYLF (accessed on 22 May 2022).

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
