# Peer review of "Estimating Mineral Requirements of Wild Herbivores: Modelling Arctic Caribou (Rangifer tarandus granti) in Summer"

_animals, 2024, doi:10.3390/ani14060868_

Round 1
Reviewer 1 Report
Comments and Suggestions for Authors
Interesting manuscript on an interesting topic. My only suggestion would be to swap around the first two paragraphs in the introduction. The end of the current paragraph 2 would flow nicely into the now paragraph 1.
Author Response
We tried rearranging the order of the paragraphs but it does not work. We have retained the structure. The first paragraph describes the minerals important for mammals. The second paragraph connects minerals in plants to herbivores. The third paragraph describes arctic caribou as an herbivore with highly seasonal supply of plant minerals.
Reviewer 2 Report
Comments and Suggestions for Authors
This research treats a critical issue in caribou's nutrition: their mineral requirements in female lactating animals. However, some details in the methodology need to be clarified.

Author Response
Title You should be more specific according to your research and include female caribou. Indicate the scientific name of arctic caribou.
RESPONSE: Title changed to include “Rangifer tarandus granti” (Line 3)
Line 69. However, this statement might not be valid when wild animals are compared to high-producing domestic animals.
RESPONSE: We have added text to line 68. “However, requirements for wild ungulates may be estimated from those of domestic ungulates by accounting for differences in seasonal food intake, mass gain and reproduction.”
We also acknowledge that requirements of domestic species are biased to young growing animals in the Discussion. Line 313 “Our estimates of mineral demands include those for growing domestic animals that are probably higher than those required for seasonal regain of fat and lean mass in adult animals such as reproductive caribou (Table 1).”
Line 98. To which domestic animals do you refer? Do you consider captive caribou? It needs to be clarified.
RESPONSE: Sentence changed to define species on Line 94. “We combined measures of intake and body mass change of captive caribou [27] with published requirements for domestic cattle (Bos taurus), goats (Capra hircus), horses (Equus ferus caballus) and sheep (Ovis aries) (Fig. 1)”.
Table 1. Why did you include dairy cows? Usually, they produce much more milk than the female caribou and some mineral requirements, such as Ca, must be higher.
RESPONSE: We have added text to line 93. “We used a factorial approach to derive mineral requirements of lactating female caribou based on the best available information.”
We did find that projections of minimum dietary Ca and Na were highest for the dairy cattle model (Fig 3 and 4). However, projections from the dairy cattle model were not the highest estimate of minimum dietary concentration for the other seven minerals (i.e., P, K, Mg, Mn, Fe, Cu, Zn). Requirements for all nine minerals have been estimated for dairy cattle but requirements for beef cattle are only available for Ca, P and Mg.
Reviewer 3 Report
Comments and Suggestions for Authors
Congratulations on your very comprehensive modelling of multiple nutritionally important minerals, a significant task well conceived and described. Good to see the relevance with which the domestic model physiologies apply to wildlife species, with ecological differences well identified and discussed.
Seriously only a couple minor typos detected: line 391 - metalloenzymes spelling and line 395 - deposition of minerals - add "of"
Author Response
RESPONSE: Corrected as suggested